# OpenSTL: A Comprehensive Benchmark of Spatio-Temporal Predictive Learning

**Cheng Tan**[1,2*]    **Siyuan Li**[1,2*]  **Zhangyang Gao**[1,2]   **Wenfei Guan**[3]   **Zedong Wang**[2]
**Zicheng Liu**[1,2]    **Lirong Wu**[1,2]    **Stan Z. Li**[2†]
[1]Zhejiang University;        [3]Xidian University;
[2]AI Lab, Research Center for Industries of the Future, Westlake University
{tancheng; lisiyuan; gaozhangyang; guanwenfei; wangzedong; liuzicheng;
wulirong; stan.zq.li}@westlake.edu.cn;

## Abstract

Spatio-temporal predictive learning is a learning paradigm that enables models to learn spatial and temporal patterns by predicting future frames from given past frames in an unsupervised manner. Despite remarkable progress in recent years, a lack of systematic understanding persists due to the diverse settings, complex implementation, and difficult reproducibility. Without standardization, comparisons can be unfair and insights inconclusive. To address this dilemma, we propose OpenSTL, a comprehensive benchmark for spatio-temporal predictive learning that categorizes prevalent approaches into recurrent-based and recurrent-free models. OpenSTL provides a modular and extensible framework implementing various state-of-the-art methods. We conduct standard evaluations on datasets across various domains, including synthetic moving object trajectory, human motion, driving scenes, traffic flow, and weather forecasting. Based on our observations, we provide a detailed analysis of how model architecture and dataset properties affect spatio-temporal predictive learning performance. Surprisingly, we find that recurrent-free models achieve a good balance between efficiency and performance than recurrent models. Thus, we further extend the common MetaFormers to boost recurrent-free spatial-temporal predictive learning. We open-source the code and models at https://github.com/chengtan9907/OpenSTL.

## 1 Introduction

Recent years have witnessed rapid and remarkable progress in spatio-temporal predictive learning [38, 29, 10, 41]. This burgeoning field aims to learn latent spatial and temporal patterns through the challenging task of forecasting future frames based solely on given past frames in an unsupervised manner [40, 55, 56, 54]. By ingesting raw sequential data, these self-supervised models [4, 14, 24] can uncover intricate spatial and temporal interdependencies without the need for tedious manual annotation, enabling them to extrapolate coherently into the future in a realistic fashion [29, 12]. Spatio-temporal predictive learning benefits a wide range of applications with its ability to anticipate the future from the past in a data-driven way, including modeling the devastating impacts of climate change [38, 35], predicting human movement [61, 45], forecasting traffic flow in transportation systems [8, 51], and learning expressive representations from video [32, 19]. By learning to predict the future without supervision from massive datasets, these techniques have the potential to transform domains where anticipation and planning are crucial but limited labeled data exists [9, 2, 44, 31].

---

[*]Equal contribution.
[†]Corresponding author.

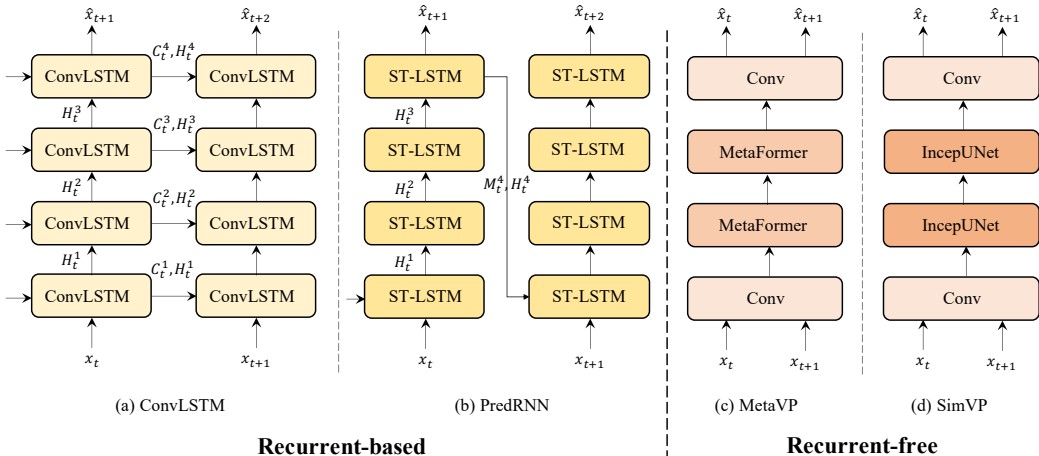

**Recurrent-based**

**Recurrent-free**

Figure 1: Two typical sptaio-temporal predictive learning models. As illustrated by the left two instances (a)(b), the first type requires several recurrent modules to predict the next frame according to the previous frames in an auto-regressive manner, dubbed recurrent-based models. As for the right two instances (c)(d), the second type predicts all future frames based on all given frames at once, which inferences in parallel and is called the recurrent-free model.

Despite the significance of spatio-temporal predictive learning and the development of various approaches, there remains a conspicuous lack of a comprehensive benchmark for this field covering various synthetic and practical application scenarios. We believe that a comprehensive benchmark is essential for advancing the field and facilitating meaningful comparisons between different methods. In particular, there exists a perennial question that has not yet been conclusively answered: *is it necessary to employ recurrent neural network architectures to capture temporal dependencies?* In other words, *can recurrent-free models achieve performance comparable to recurrent-based models without explicit temporal modeling*?

Since the seminal work ConvLSTM [38] was proposed, which ingeniously integrates convolutional networks and long-short term memory (LSTM) networks [15] to separately capture spatial and temporal correlations, researchers have vacillated between utilizing or eschewing recurrent architectures. As shown in Figure 1, (a) ConvLSTM is a prototypical recurrent-based model that infuses a recurrent structure into convolutional networks. (b) PredRNN [49] represents a series of recurrent models that revise the flow of information to enhance performance. (c) MetaVP is the recurrent-free model that abstracted from SimVP by substituting its IncepU [10] modules with MetaFormers [59]. (d) SimVP [10, 40] is a typical recurrent-free model that achieves performance comparable to previous state-of-the-art models without explicitly modeling temporal dependencies.

In this study, we illuminate the long-standing question of whether explicit temporal modeling with recurrent neural networks is requisite for spatio-temporal predictive learning. To achieve this, we present a comprehensive benchmark, **Open S**patio-**T**emporal predictive **L**earning, dubbed OpenSTL. We revisit the approaches that represent the foremost strands within a modular and extensive framework to ensure fair comparisons. We summarize our main contributions as follows:

- We build OpenSTL, a comprehensive benchmark for spatio-temporal predictive learning that includes 14 representative algorithms and 24 models. OpenSTL covers a wide range of methods and classifies them into two categories: recurrent-based and recurrent-free methods.

- We conduct extensive experiments on a diversity of tasks ranging from synthetic moving object trajectories to real-world human motion, driving scenes, traffic flow, and weather forecasting. The datasets span synthetic to real-world data and micro-to-macro scales.

- While recurrent-based models have been well developed, we rethink the potential of recurrent-free models based on insights from OpenSTL. We propose generalizing MetaFormer-like architectures [59] to boost recurrent-free spatio-temporal predictive learning. Recurrent-free models can thus reformulate the problem as a downstream task of designing vision backbones for general applications.

## 2 Background and Related work

### 2.1 Problem definition

We propose the formal definition for the spatio-temporal predictive learning problem as follows. Given a sequence of video frames $\mathcal{X}^{t,T} = \{\boldsymbol{x}^i\}_{t-T+1}^t$ up to time $t$ spanning the past $T$ frames, the objective is to predict the subsequent $T'$ frames $\mathcal{Y}^{t+1,T'} = \{\boldsymbol{x}^i\}_{t+1}^{t+1+T'}$ from time $t+1$ onwards, where each frame $\boldsymbol{x}_i \in \mathbb{R}^{C \times H \times W}$ typically comprises $C$ channels, with height $H$ and width $W$ pixels. In practice, we represent the input sequence of observed frames and output sequence of predicted frames respectively as tensors $\mathcal{X}^{t,T} \in \mathbb{R}^{T \times C \times H \times W}$ and $\mathcal{Y}^{t+1,T'} \in \mathbb{R}^{T' \times C \times H \times W}$.

The model with learnable parameters $\Theta$ learns a mapping $\mathcal{F}_\Theta : \mathcal{X}^{t,T} \mapsto \mathcal{Y}^{t+1,T'}$ by leveraging both spatial and temporal dependencies. In our case, the mapping $\mathcal{F}_\Theta$ corresponds to a neural network trained to minimize the discrepancy between the predicted future frames and the ground-truth future frames. The optimal parameters $\Theta^*$ are given by:

$$\Theta^* = \arg\min_\Theta \mathcal{L}(\mathcal{F}_\Theta(\mathcal{X}^{t,T}), \mathcal{Y}^{t+1,T'}), \tag{1}$$

where $\mathcal{L}$ denotes a loss function that quantifies such discrepancy.

In this study, we categorize prevalent spatio-temporal predictive learning methods into two classes: recurrent-based and recurrent-free models. For *recurrent-based models*, the mapping $\mathcal{F}_\Theta$ comprises several recurrent interactions:

$$\mathcal{F}_\Theta : f_\theta(\boldsymbol{x}^{t-T+1}, \boldsymbol{h}^{t-T+1}) \circ ... \circ f_\theta(\boldsymbol{x}^t, \boldsymbol{h}^t) \circ ... \circ f_\theta(\boldsymbol{x}^{t+T'-1}, \boldsymbol{h}^{t+T'-1}), \tag{2}$$

where $\boldsymbol{h}^i$ represents the memory state encompassing historical information and $f_\theta$ denotes the mapping between each pair of adjacent frames. The parameters $\theta$ are shared across each state. Therefore, the prediction process can be expressed as follows:

$$\boldsymbol{x}^{t+1} = f_\theta(\boldsymbol{x}^i, \boldsymbol{h}^i), \forall i \in \{t+1, \cdots, t+T'\}, \tag{3}$$

For *recurrent-free* models, the prediction process directly feeds the whole sequence of observed frames into the model and outputs the complete predicted frames at once.

### 2.2 Recurrent-based models

Since the pioneering work ConvLSTM [38] was proposed, recurrent-based models [29, 30, 16, 12, 58, 31] have been extensively studied. PredRNN [49] adopts vanilla ConvLSTM modules to build a Spatio-temporal LSTM (ST-LSTM) unit that models spatial and temporal variations simultaneously. PredRNN++ [47] proposes a gradient highway unit to mitigate the gradient vanishing and a Casual-LSTM module to cascadely connect spatial and temporal memories. PredRNNv2 [50] further proposes a curriculum learning strategy and a memory decoupling loss to boost performance. MIM [51] introduces high-order non-stationarity learning in designing LSTM modules. PhyDNet [12] explicitly disentangles PDE dynamics from unknown complementary information with a recurrent physical unit. E3DLSTM [48] integrates 3D convolutions into recurrent networks. MAU [3] proposes a motion-aware unit that captures motion information. Although various recurrent-based models have been developed, the reasons behind their strong performance remain not fully understood.

### 2.3 Recurrent-free models

Compared to recurrent-based models, recurrent-free models have received less attention. Previous studies tend to use 3D convolutional networks to model temporal dependencies [28, 1]. PredCNN [57] and TrajectoryCNN [25] use 2D convolutional networks for efficiency. However, early recurrent-free models were doubted due to their poor performance. Recently, SimVP [10, 40, 41] provided a simple but effective recurrent-free baseline with competitive performance. PastNet [53] and IAM4VP [37] are two recent recurrent-free models that perform strong performance. In this study, we implemented representative recurrent-based and recurrent-free models under a unified framework to systematically investigate their intrinsic properties. Moreover, we further explored the potential of recurrent-free models by reformulating the spatio-temporal predictive learning problem and extending MetaFormers [59] to bridge the gap between the visual backbone and spatio-temporal learning.

# 3 OpenSTL

## 3.1 Supported Methods

### 3.1.1 Overview

OpenSTL has implemented 14 representative spatio-temporal predictive learning methods under a unified framework, including 11 recurrent-based methods and 3 recurrent-free methods. We summarize these methods in Table 1, where we also provide the corresponding conference/journal and the types of their spatial-temporal modeling components. The spatial modeling of these methods is fundamentally consistent. Most methods apply two-dimensional convolutional networks (Conv2D) to model spatial dependencies, while E3D-LSTM and CrevNet harness three-dimensional convolutional networks (Conv3D) instead.

The primary distinction between these methods lies in how they model temporal dependencies using their proposed modules. The ST-LSTM module, proposed in PredRNN [49], is the most widely used module. CrevNet has a similar modeling approach as PredRNN, but it incorporates an information-preserving mechanism into the model. Analogously, Casual-LSTM [47], MIM Block [51], E3D-LSTM [48], PhyCell [12], and MAU [3] represent variants of ConvLSTM proposed with miscellaneous motivations. MVFB is built as a multi-scale voxel flow block that diverges from ConvLSTM. However, DMVFN [17] predicts future frames frame-by-frame which still qualifies as a recurrent-based model. IncepU [10] constitutes an Unet-like module that also exploits the multi-scale feature from the InceptionNet-like architecture. gSTA [40] and TAU [41] extend the IncepU module to simpler and more efficient architectures without InceptionNet or Unet-like architectures. In this work, we further extend the temporal modeling of recurrent-free models by introducing MetaFormers [59] to boost recurrent-free spatio-temporal predictive learning.

Table 1: Categorizations of the supported spatial-temporal predictive learning methods in OpenSTL.

| Category | Method | Conference/Journal | Spatial modeling | Temporal modeling |
|---|---|---|---|---|
| Recurrent-based | ConvLSTM [38] | NeurIPS 2015 | Conv2D | Conv-LSTM |
| | PredNet [29] | ICLR 2017 | Conv2D | ST-LSTM |
| | PredRNN [49] | NeurIPS 2017 | Conv2D | ST-LSTM |
| | PredRNN++ [47] | ICML 2018 | Conv2D | Casual-LSTM |
| | MIM [51] | CVPR 2019 | Conv2D | MIM Block |
| | E3D-LSTM [48] | ICLR 2019 | Conv3D | E3D-LSTM |
| | CrevNet [58] | ICLR 2020 | Conv3D | ST-LSTM |
| | PhyDNet [12] | CVPR 2020 | Conv2D | ConvLSTM+PhyCell |
| | MAU [3] | NeurIPS 2021 | Conv2D | MAU |
| | PredRNNv2 [50] | TPAMI 2022 | Conv2D | ST-LSTM |
| | DMVFN [17] | CVPR 2023 | Conv2D | MVFB |
| Recurrent-free | SimVP [10] | CVPR 2022 | Conv2D | IncepU |
| | TAU [41] | CVPR 2023 | Conv2D | TAU |
| | SimVPv2 [40] | arXiv | Conv2D | gSTA |

### 3.1.2 Rethink the recurrent-free models

Although less studied, recurrent-free spatio-temporal predictive learning models share a similar architecture, as illustrated in Figure 2. The encoder comprises several 2D convolutional networks, which project high-dimensional input data into a low-dimensional latent space. When given a batch of input observed frames $\mathcal{B} \in \mathbb{R}^{B \times T \times C \times H \times W}$, the encoder focuses solely on intra-frame spatial correlations, ignoring temporal modeling. Subsequently, the middle temporal module stacks the low-dimensional representations along the temporal dimension to ascertain temporal dependencies. Finally, the decoder comprises several 2D convolutional upsampling networks, which reconstruct subsequent frames from the learned latent representations.

The encoder and decoder enable efficient temporal learning by modeling temporal dependencies in a low-dimensional latent space. The core component of recurrent-free models is the temporal module. Previous studies have proposed temporal modules such as IncepU [10], TAU [41], and

gSTA [40] that have proved beneficial. However, we argue that the competence stems primarily from the general recurrent-free architecture instead of the specific temporal modules. Thus, we employ MetaFormers [59] as the temporal module by changing the input channels from the original $C$ to inter-frame channels $T \times C$. By extending the recurrent-free architecture, we leverage the advantages of MetaFormers to enhance the recurrent-free model. In this work, we implement ViT [7], Swin Transformer [26], Uniformer [21], MLP-Mixer [42], ConvMixer [43], Poolformer [59], ConvNeXt [27], VAN [13], HorNet [33], and MogaNet [22] for the MetaFormers-based recurrent-free model, substituting the intermediate temporal module in the original recurrent-free architecture.

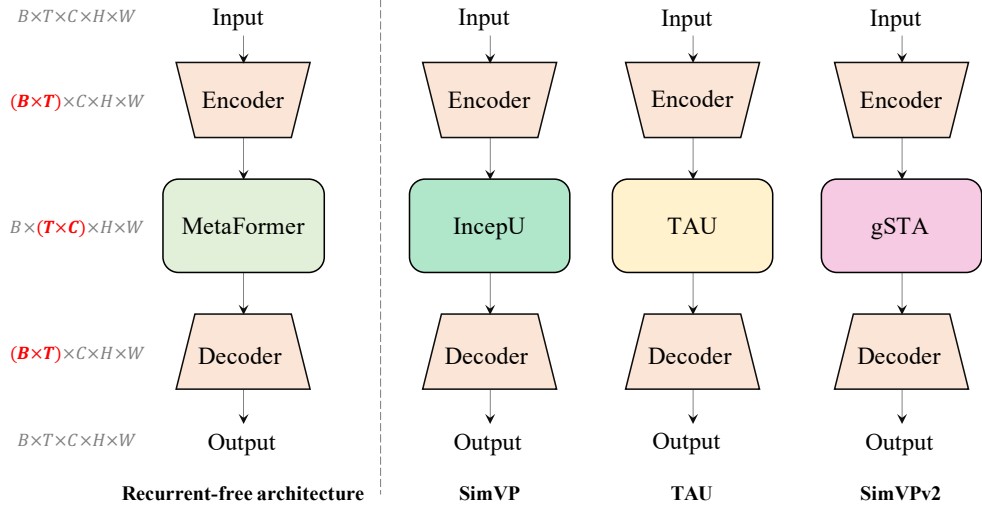

Figure 2: The general architecture of recurrent-free models with three instances.

## 3.2 Supported Tasks

We have curated five diverse tasks in our OpenSTL benchmark, which cover a wide range of scenarios from synthetic simulations to real-world situations at various scales. The tasks include synthetic moving object trajectories, real-world human motion capture, driving scenes, traffic flow, and weather forecasting. The datasets used in our benchmark range from synthetic to real-world, and from micro to macro scales. We have provided a summary of the dataset statistics in Table 2.

Table 2: The detailed dataset statistics of the supported tasks in OpenSTL.

| Dataset | Training size | Testing size | Channel | Height | Width | $T$ | $T'$ |
|---|---|---|---|---|---|---|---|
| Moving MNIST variants | 10,000 | 10,000 | 1 / 3 | 64 | 64 | 10 | 10 |
| KTH Action | 4,940 | 3,030 | 1 | 128 | 128 | 10 | 20/40 |
| Human3.6M | 73,404 | 8,582 | 3 | 128 | 128 | 4 | 4 |
| Kitti&Caltech | 3,160 | 3,095 | 3 | 128 | 160 | 10 | 1 |
| TaxiBJ | 20,461 | 500 | 2 | 32 | 32 | 4 | 4 |
| WeatherBench-S | 2,167 | 706 | 1 | 32/128 | 64/256 | 12 | 12 |
| WeatherBench-M | 54,019 | 2,883 | 4 | 32 | 64 | 4 | 4 |

**Synthetic moving object trajectory prediction** *Moving MNIST* [39] is one of the seminal benchmark datasets that has been extensively utilized. Each video sequence comprises two moving digits confined within a $64 \times 64$ frame. Each digit was assigned a velocity whose direction was randomly chosen from a unit circle and whose magnitude was also arbitrarily selected from a fixed range. Apart from the original Moving MNIST dataset, we provide two variants with more complicated objects (*Moving FashionMNIST*) that replace the digits with fashion objects and more complex scenes (*Moving MNIST-CIFAR*) that employ images from the CIFAR-10 dataset [20] as the background. Moreover, we provide three settings of Moving MNIST for robustness evaluations, including missing frames, dynamic noise, and perceptual occlusions.

**Human motion capture** Predicting human motion is challenging due to the complexity of human movements, which vary greatly among individuals and actions. We utilized the *KTH* dataset [36], which includes six types of human actions: walking, jogging, running, boxing, hand waving, and hand clapping. We furnish two settings, predicting the next 20 and 40 frames respectively. *Human3.6M* [18] is an intricate human pose dataset containing high-resolution RGB videos. Analogous to preceding studies [12, 51], we predict the next four frames by the observed four frames.

**Driving scene prediction** Predicting the future dynamics of driving scenarios is crucial for autonomous driving. Compared to other tasks, this undertaking involves non-stationary and diverse scenes. To address this issue, we follow the conventional approach [29] and train the model on the *Kitti* [11] dataset. We then evaluate the performance on the *Caltech Pedestrian* [6] dataset. To ensure consistency, we center-cropped and downsized all frames to $128 \times 160$ pixels.

**Traffic flow prediction** Forecasting the dynamics of crowds is crucial for traffic management and public safety. To evaluate spatio-temporal predictive learning approaches for traffic flow prediction, we use the *TaxiBJ* [60] dataset. This dataset includes GPS data from taxis and meteorological data in Beijing. The dataset contains two types of crowd flows, representing inflow and outflow. The temporal interval is 30 minutes, and the spatial resolution is $32 \times 32$.

**Weather forecasting** Global weather pattern prediction is an essential natural predicament. The WeatherBench [34] dataset is a large-scale weather forecasting dataset encompassing various types of climatic factors. The raw data is re-grid to $5.625°$ resolution ($32 \times 64$ grid points) and $1.40625°$ ($128 \times 256$ grid points). We consider two setups: First, *WeatherBench-S* is a single-variable setup in which each climatic factor is trained independently. The model is trained on data from 2010-2015, validated on data from 2016, and tested on data from 2017-2018, with a one-hour temporal interval. Second, *WeatherBench-M* is a multi-variable setup that mimics real-world weather forecasting more closely. All climatic factors are trained simultaneously. The model is trained on data from 1979 to 2015, using the same validation and testing data as WeatherBench-S. The temporal interval is extended to six hours, capturing a broader range of temporal dependencies.

## 3.3 Evaluation Metrics

We evaluate the performance of supported models on the aforementioned tasks using various metrics in a thorough and rigorous manner. We use them for specific tasks according to their characteristics.

**Error metrics** We utilize the mean squared error (MSE) and mean absolute error (MAE) to evaluate the difference between the predicted results and the true targets. Root mean squared error (RMSE) is also used in weather forecasting as it is more common in this domain.

**Similarity metrics** We utilize the structural similarity index measure (SSIM) [52] and peak signal-to-noise ratio (PSNR) to evaluate the similarity between the predicted results and the true targets. Such metrics are extensively used in image processing and computer vision.

**Perceptual metrics** LPIPS [62] is implemented to evaluate the perceptual difference between the predicted results and the true targets in the human visual system. LPIPS provides a perceptually-aligned evaluation for vision tasks. We utilize this metric in real-world video prediction tasks.

**Computational metrics** We utilize the number of parameters and the number of floating-point operations (FLOPs) to evaluate the computational complexity of the models. We also report the frames per second (FPS) on a single NVIDIA V100 GPU to evaluate the inference speed.

## 3.4 Codebase Structure

While existing open-sourced spatio-temporal predictive learning codebases are independent, OpenSTL provides a modular and extensible framework that adheres to the design principles of OpenMMLab [5] and assimilates code elements from OpenMixup [23] and USB [46]. OpenSTL excels in user-friendliness, organization, and comprehensiveness, surpassing the usability of existing open-source STL codebases. A detailed description of the codebase structure can be found in Appendix B.

# 4 Experiment and Analysis

We conducted comprehensive experiments on the mentioned tasks to assess the performance of the supported methods in OpenSTL. Detailed analysis of the results is presented to gain insights into spatio-temporal predictive learning. Implementation details can be found in Appendix C.

## 4.1 Synthetic Moving Object Trajectory Prediction

We conduct experiments on the synthetic moving object trajectory prediction task, utilizing three datasets: Moving MNIST, Moving FashionMNIST, and Moving MNIST-CIFAR. The performance of the evaluated models on the Moving MNIST dataset is reported in Table 3. The detailed results for the other two synthetic datasets are in Appendix D.1.

It can be observed that recurrent-based models yield varied results that do not consistently outperform recurrent-free models, while recurrent-based models always exhibit slower inference speeds than their recurrent-free counterparts. Although PredRNN, PredRNN++, MIM, and PredRNNv2 achieve lower MSE and MAE values compared to recurrent-free models, their FLOPs are nearly five times higher, and their FPS are approximately seven times slower than all recurrent-free models. Furthermore, there are minimal disparities in the performance of recurrent-free models as opposed to recurrent-based models, highlighting the robustness of the proposed general recurrent-free architecture. The remaining two synthetic datasets, consisting of more intricate moving objects (Moving FashionMNIST) and complex scenes (Moving MNIST-CIFAR), reinforce the experimental findings that recurrent-free models deliver comparable performance with significantly higher efficiency. In these toy datasets characterized by high frequency but low resolution, recurrent-based models excel in capturing temporal dependencies but are susceptible to high computational complexity.

Table 3: The performance on the Moving MNIST dataset.

| | Method | Params (M) | FLOPs (G) | FPS | MSE ↓ | MAE ↓ | SSIM ↑ | PSNR ↑ |
|---|---|---|---|---|---|---|---|---|
| Recurrent-based | ConvLSTM | 15.0 | 56.8 | 113 | 29.80 | 90.64 | 0.9288 | 22.10 |
| | PredNet | 12.5 | 8.4 | 659 | 161.38 | 201.16 | 0.7783 | 14.67 |
| | PredRNN | 23.8 | 116.0 | 54 | 23.97 | 72.82 | 0.9462 | 23.28 |
| | PredRNN++ | 38.6 | 171.7 | 38 | **22.06** | **69.58** | **0.9509** | **23.65** |
| | MIM | 38.0 | 179.2 | 37 | 22.55 | 69.97 | 0.9498 | 23.56 |
| | E3D-LSTM | 51.0 | 298.9 | 18 | 35.97 | 78.28 | 0.9320 | 21.11 |
| | CrevNet | 5.0 | 270.7 | 10 | 30.15 | 86.28 | 0.9350 | 22.15 |
| | PhyDNet | 3.1 | 15.3 | 182 | 28.19 | 78.64 | 0.9374 | 22.62 |
| | MAU | 4.5 | 17.8 | 201 | 26.86 | 78.22 | 0.9398 | 22.57 |
| | PredRNNv2 | 23.9 | 116.6 | 52 | 24.13 | 73.73 | 0.9453 | 23.21 |
| | DMVFN | 3.5 | 0.2 | 1145 | 123.67 | 179.96 | 0.8140 | 16.15 |
| Recurrent-free | SimVP | 58.0 | 19.4 | 209 | 32.15 | 89.05 | 0.9268 | 21.84 |
| | TAU | 44.7 | 16.0 | 283 | 24.60 | 71.93 | 0.9454 | 23.19 |
| | SimVPv2 | 46.8 | 16.5 | 282 | 26.69 | 77.19 | 0.9402 | 22.78 |
| | ViT | 46.1 | 16.9 | 290 | 35.15 | 95.87 | 0.9139 | 21.67 |
| | Swin Transformer | 46.1 | 16.4 | 294 | 29.70 | 84.05 | 0.9331 | 22.22 |
| | Uniformer | 44.8 | 16.5 | 296 | 30.38 | 85.87 | 0.9308 | 22.13 |
| | MLP-Mixer | 38.2 | 14.7 | 334 | 29.52 | 83.36 | 0.9338 | 22.22 |
| | ConvMixer | 3.9 | 5.5 | 658 | 32.09 | 88.93 | 0.9259 | 21.93 |
| | Poolformer | 37.1 | 14.1 | 341 | 31.79 | 88.48 | 0.9271 | 22.03 |
| | ConvNext | 37.3 | 14.1 | 344 | 26.94 | 77.23 | 0.9397 | 22.74 |
| | VAN | 44.5 | 16.0 | 288 | 26.10 | 76.11 | 0.9417 | 22.89 |
| | HorNet | 45.7 | 16.3 | 287 | 29.64 | 83.26 | 0.9331 | 22.26 |
| | MogaNet | 46.8 | 16.5 | 255 | 25.57 | 75.19 | 0.9429 | 22.99 |

## 4.2 Real-world Video Prediction

We perform experiments on real-world video predictions, specifically focusing on human motion capturing using the KTH and Human3.6M datasets, as well as driving scene prediction using the Kitti&Caltech dataset. Due to space constraints, we present the results for the Kitti&Caltech dataset in Table 4, while the detailed results for the other datasets can be found in Appendix D.2. We observed that as the resolution increases, the computational complexity of recurrent-based models dramatically increases. In contrast, recurrent-free models achieve a commendable balance between efficiency and performance. Notably, although some recurrent-based models achieve lower MSE and MAE values, their FLOPs are nearly 20 times higher compared to their recurrent-free counterparts. This highlights the efficiency advantage of recurrent-free models, especially in high-resolution scenarios.

Table 4: The performance on the Kitti&Caltech dataset.

| Method | | Params (M) | FLOPs (G) | FPS | MSE ↓ | MAE ↓ | SSIM ↑ | PSNR ↑ | LPIPS ↓ |
|---|---|---|---|---|---|---|---|---|---|
| Recurrent-based | ConvLSTM | 15.0 | 595.0 | 33 | 139.6 | 1583.3 | 0.9345 | 27.46 | 8.58 |
| | PredNet | 12.5 | 42.8 | 94 | 159.8 | 1568.9 | 0.9286 | 27.21 | 11.29 |
| | PredRNN | 23.7 | 1216.0 | 17 | 130.4 | 1525.5 | 0.9374 | 27.81 | 7.40 |
| | PredRNN++ | 38.5 | 1803.0 | 12 | **125.5** | **1453.2** | 0.9433 | 28.02 | 13.21 |
| | MIM | 49.2 | 1858.0 | 39 | 125.1 | 1464.0 | 0.9409 | **28.10** | 6.35 |
| | E3D-LSTM | 54.9 | 1004.0 | 10 | 200.6 | 1946.2 | 0.9047 | 25.45 | 12.60 |
| | PhyDNet | 3.10 | 40.4 | 117 | 312.2 | 2754.8 | 0.8615 | 23.26 | 32.19 |
| | MAU | 24.3 | 172.0 | 16 | 177.8 | 1800.4 | 0.9176 | 26.14 | 9.67 |
| | PredRNNv2 | 23.8 | 1223.0 | 16 | 147.8 | 1610.5 | 0.9330 | 27.12 | 8.92 |
| | DMVFN | 3.6 | 1.2 | 557 | 183.9 | 1531.1 | 0.9314 | 26.78 | **4.94** |
| Recurrent-free | SimVP | 8.6 | 60.6 | 57 | 160.2 | 1690.8 | 0.9338 | 26.81 | 6.76 |
| | TAU | 15.0 | 92.5 | 55 | 131.1 | 1507.8 | 0.9456 | 27.83 | 5.49 |
| | SimVPv2 | 15.6 | 96.3 | 40 | 129.7 | 1507.7 | 0.9454 | 27.89 | 5.57 |
| | ViT | 12.7 | 155.0 | 25 | 146.4 | 1615.8 | 0.9379 | 27.43 | 6.66 |
| | Swin Transformer | 15.3 | 95.2 | 49 | 155.2 | 1588.9 | 0.9299 | 27.25 | 8.11 |
| | Uniformer | 11.8 | 104.0 | 28 | 135.9 | 1534.2 | 0.9393 | 27.66 | 6.87 |
| | MLP-Mixer | 22.2 | 83.5 | 60 | 207.9 | 1835.9 | 0.9133 | 26.29 | 7.75 |
| | ConvMixer | 1.5 | 23.1 | 129 | 174.7 | 1854.3 | 0.9232 | 26.23 | 7.76 |
| | Poolformer | 12.4 | 79.8 | 51 | 153.4 | 1613.5 | 0.9334 | 27.38 | 7.00 |
| | ConvNext | 12.5 | 80.2 | 54 | 146.8 | 1630.0 | 0.9336 | 27.19 | 6.99 |
| | VAN | 14.9 | 92.5 | 41 | 127.5 | 1476.5 | 0.9462 | 27.98 | 5.50 |
| | HorNet | 15.3 | 94.4 | 43 | 152.8 | 1637.9 | 0.9365 | 27.09 | 6.00 |
| | MogaNet | 15.6 | 96.2 | 36 | 131.4 | 1512.1 | **0.9442** | 27.79 | 5.39 |

## 4.3 Traffic and Weather Forecasting

Traffic flow prediction and weather forecasting are two critical tasks that have significant implications for public safety and scientific research. While these tasks operate at a macro level, they exhibit lower frequencies compared to the tasks mentioned above, and the states along the timeline tend to be more stable. Capturing subtle changes in such tasks poses a significant challenge. In order to assess the performance of the supported models in OpenSTL, we conduct experiments on the TaxiBJ and WeatherBench datasets. It is worth noting that weather forecasting encompasses various settings, and we provide detailed results of them in Appendix D.3.

Here, we present a comparison of the MAE and RMSE metrics for representative approaches in single-variable weather factor forecasting at low resolution. Figure 3 displays the results for four climatic factors, i.e., temperature, humidity, wind component, and cloud cover. Notably, recurrent-free models consistently outperform recurrent-based models across all weather factors, indicating their potential to apply spatio-temporal predictive learning to macro-scale tasks instead of relying solely on recurrent-based models. These findings underscore the promising nature of recurrent-free models and suggest that they can be a viable alternative to the prevailing recurrent-based models in

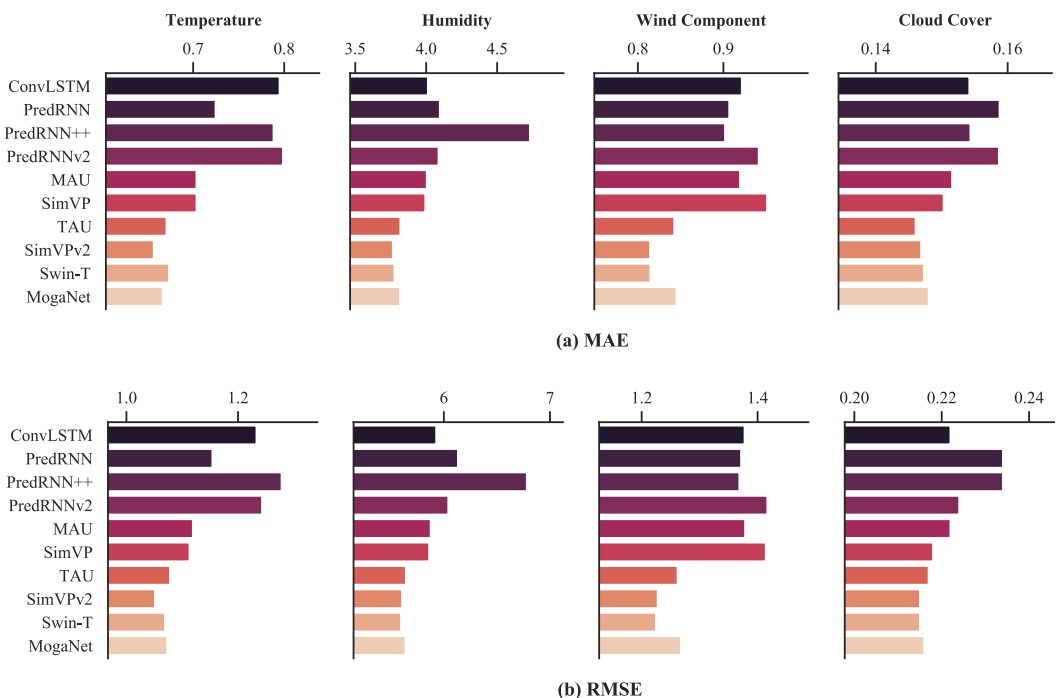

**(a) MAE**

**(b) RMSE**

Figure 3: The (a) MAE and (b) RMSE metrics of the representative approaches on the four weather forecasting tasks in WeatherBench.

the context of weather forecasting. Furthermore, in the Appendix, we provide additional insights into high-resolution and multi-variable weather forecasting, where similar trends are observed.

## 4.4 Robustness Analysis

To further understand the differences in robustness between the recurrent-based and recurrent-free spatiotemporal predictive learning methods, we constructed three experimental setups: (i) Moving MNIST - Missing, which deals with input frames with missing frames, where we set the probability of random missing frame to 20%; (ii) Moving MNIST - Dynamic, where we added random Gaussian noise to the speed of each digit, making their movement speeds irregular; (iii) Moving MNIST - Perceptual, where we randomly occluded the input frames, using a black 24×24 patch for occlusion. We choose three representative recurrent-based and three recurrent-free methods for evaluation. The experimental results for these three setups are presented in Table 5, 6, 7, respectively.

It can be observed that the recurrent-free methods exhibit remarkable robustness under both the missing and perceptual noise scenarios. Even when compared to situations without noise, there is little performance degradation due to their focus on global information. Conversely, recurrent-based methods encounter substantial performance drops. They overly focus on the relationships individual frames can inadvertently lead to overfitting. In the case of dynamic noise, all methods faced significant performance setbacks, because the speed of the digits became irregular and harder to predict.

Table 5: The performance on the Moving MNIST - Missing dataset.

| Method | | Params (M) | FLOPs (G) | FPS | MSE ↓ | MAE ↓ | SSIM ↑ | PSNR ↑ |
|---|---|---|---|---|---|---|---|---|
| | ConvLSTM | 15.0 | 56.8 | 113 | 32.73 | 96.95 | 0.9201 | 21.65 |
| Recurrent-based | PredRNN | 23.8 | 116.0 | 54 | 46.05 | 117.21 | 0.8800 | 20.35 |
| | PredRNN++ | 38.6 | 171.7 | 38 | 53.89 | 118.45 | 0.8907 | 19.71 |
| | SimVP | 58.0 | 19.4 | 209 | 34.92 | 95.23 | 0.9194 | 21.44 |
| Recurrent-free | TAU | 44.7 | 16.0 | 283 | **26.77** | **77.50** | **0.9400** | **22.74** |
| | SimVPv2 | 46.8 | 16.5 | 282 | 28.63 | 81.79 | 0.9352 | 22.39 |

Table 6: The performance on the Moving MNIST - Dynamic dataset.

| Method | | Params (M) | FLOPs (G) | FPS | MSE ↓ | MAE ↓ | SSIM ↑ | PSNR ↑ |
|---|---|---|---|---|---|---|---|---|
| Recurrent-based | ConvLSTM | 15.0 | 56.8 | 113 | 49.03 | 135.49 | 0.8683 | 19.73 |
| | PredRNN | 23.8 | 116.0 | 54 | 59.18 | 157.47 | 0.8220 | 19.09 |
| | PredRNN++ | 38.6 | 171.7 | 38 | **40.85** | **109.32** | **0.9030** | **20.65** |
| Recurrent-free | SimVP | 58.0 | 19.4 | 209 | 48.41 | 130.83 | 0.8725 | 19.91 |
| | TAU | 44.7 | 16.0 | 283 | 43.37 | 121.31 | 0.8853 | 20.41 |
| | SimVPv2 | 46.8 | 16.5 | 282 | 44.74 | 123.70 | 0.8823 | 20.28 |

Table 7: The performance on the Moving MNIST - Perceptual dataset.

| Method | | Params (M) | FLOPs (G) | FPS | MSE ↓ | MAE ↓ | SSIM ↑ | PSNR ↑ |
|---|---|---|---|---|---|---|---|---|
| Recurrent-based | ConvLSTM | 15.0 | 56.8 | 113 | 31.34 | 95.39 | 0.9227 | 21.85 |
| | PredRNN | 23.8 | 116.0 | 54 | 46.04 | 122.40 | 0.8792 | 20.28 |
| | PredRNN++ | 38.6 | 171.7 | 38 | 51.76 | 127.12 | 0.8722 | 19.85 |
| Recurrent-free | SimVP | 58.0 | 19.4 | 209 | 34.73 | 95.23 | 0.9196 | 21.44 |
| | TAU | 44.7 | 16.0 | 283 | **26.87** | **78.08** | **0.9393** | **22.69** |
| | SimVPv2 | 46.8 | 16.5 | 282 | 28.83 | 82.65 | 0.9343 | 22.36 |

## 5 Conclusion and Discussion

This paper introduces OpenSTL, a comprehensive benchmark for spatio-temporal predictive learning with a diverse set of 14 representative methods and 24 models, addressing a wide range of challenging tasks. OpenSTL categorizes existing approaches into recurrent-based and recurrent-free models. To unlock the potential of recurrent-free models, we propose a general recurrent-free architecture and introduce MetaFormers for temporal modeling. Extensive experiments are conducted to systematically evaluate the performance of the supported models across various tasks. In synthetic datasets, recurrent-based models excel at capturing temporal dependencies, while recurrent-free models achieve comparable performance with significantly higher efficiency. In real-world video prediction tasks, recurrent-free models strike a commendable balance between efficiency and performance. Additionally, recurrent-free models demonstrate significant superiority over their counterparts in weather forecasting, highlighting their potential for scientific applications at a macro-scale level.

Moreover, we observed that *recurrent architectures are beneficial in capturing temporal dependencies, but they are not always necessary, especially for computationally expensive tasks*. Recurrent-free models can be a viable alternative that provides a good balance between efficiency and performance. The effectiveness of recurrent-based models in capturing high-frequency spatio-temporal dependencies can be attributed to their sequential tracking of frame-by-frame changes, providing a local temporal inductive bias. On the other hand, recurrent-free models combine multiple frames together, exhibiting a global temporal inductive bias that is suitable for low-frequency spatio-temporal dependencies. We hope that our work provides valuable insights and serves as a reference for future research.

While our primary focus lies in general spatio-temporal predictive learning, there are still several open problems that require further investigation. One particular challenge is finding ways to effectively leverage the strengths of both recurrent-based and recurrent-free models to enhance the modeling of spatial-temporal dependencies. While there is a correspondence between the spatial encoding and temporal modeling in MetaVP and the token mixing and channel mixing in MetaFormer, it raises the question of whether we can improve recurrent-free models by extending the existing MetaFormers.

## Acknowledgments and Disclosure of Funding

This work was supported by the National Key R&D Program of China (2022ZD0115100), the National Natural Science Foundation of China (U21A20427), the Competitive Research Fund (WU2022A009) from the Westlake Center for Synthetic Biology and Integrated Bioengineering.

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
