# OpenReview forum: "OpenSTL: A Comprehensive Benchmark of Spatio-Temporal Predictive Learning"
_NeurIPS.cc/2023/Track/Datasets_and_Benchmarks — NeurIPS 2023 Datasets and Benchmarks Poster_

### Official Review · Reviewer_sUSg · 2023-06-23

**Rating:** 6
**Confidence:** 3
**Correctness:** Yes
**Clarity:** Yes

**Strengths:**

- The task of predictive spatio-temporal learning is important, and the community lacks a unified benchmark. Therefore, OpenSTL can facilitate the reproducibility of the field. This will benefit the researchers in the STL community
- The code has been released with a clear structure and detailed documents. I appreciate the authors' efforts in incorporating so many algorithms and datasets into one codebase
- The observation on recurrent-free methods is interesting, especially on its inference speed and computational requirements

**Additional Feedback:**

It's interesting (and somewhat sad) to see that PredRNN++, a method proposed in 2018, is among the best methods in micro-scale tasks, such as moving-MNIST, Kitti&Caltech. In contrast, on macro-scale datasets, PredRNN++ seems to perform much worse, even worse than other recurrent-based methods most of the time as shown in Figure 3. Do the authors have insight in this performance gap? Moreover, what is the biggest difference between micro-scale and macro-scale tasks? It seems to me that they require different aspects of model capacity.

**Documentation:**

Yes

**Limitations:**

The paper only briefly talks about the future directions of predictive spatio-temporal learning in the last paragraph of the conclusion section. I'd encourage the authors to discuss more the limitations of the current benchmark.

I don't see any big potential negative societal impact of the paper.

**Opportunities For Improvement:**

I only see two minor issues with the paper:
1. Missing an interesting type of video prediction datasets: BAIR [1], RoboNet [2]. The included datasets already span a wide range of tasks. But it would be interesting to see these robotics-related datasets being included.

[1] Frederik Ebert, Chelsea Finn, Alex X Lee, and Sergey Levine. Self-supervised visual planning with temporal skip connections. CoRL. 2017.

[2] Dasari, Sudeep, Frederik Ebert, Stephen Tian, Suraj Nair, Bernadette Bucher, Karl Schmeckpeper, Siddharth Singh, Sergey Levine, and Chelsea Finn. RoboNet: Large-scale multi-robot learning. CoRL. 2019.

2. The integration of recurrent-free models and MetaFormers is inspiring. The authors test a wide range of popular Transformer architectures. However, I do not see a consistent trend in the performance between these models. Can the authors provide more analysis here? E.g., if I am a beginner in STL, I'd like to know what temporal module should I choose for a recurrent-free predictor


**Relation To Prior Work:**

Yes

**Summary And Contributions:**

This paper introduces a benchmark named OpenSTL for spatio-temporal predictive learning. OpenSTL implements several classic and state-of-the-art methods, and supports several common STL datasets, including synthetic/natural video prediction, traffic flow estimation, and weather forecasting. In addition, the paper highlights recent advances in recurrent-free models. To further study their effectiveness, the authors employ MetaFormers as their middle temporal module.

Extensive benchmark results are presented in the paper. While the traditional recurrent-based methods are good at video prediction, the recurrent-free methods excel in micro-scale tasks such as weather forecasting, and require much less computation.

---

> ### Author Response · Authors · 2023-08-17
>
> Dear Reviewer sUSg,
>
> Thank you for your thoughtful and inspiring comment!
>
> ***
>
> **Q1** Missing an interesting type of video prediction datasets: BAIR [1], RoboNet [2]. The included datasets already span a wide range of tasks. But it would be interesting to see these robotics-related datasets being included.
>
> [1] Frederik Ebert, Chelsea Finn, Alex X Lee, and Sergey Levine. Self-supervised visual planning with temporal skip connections. CoRL. 2017.
>
> [2] Dasari, Sudeep, Frederik Ebert, Stephen Tian, Suraj Nair, Bernadette Bucher, Karl Schmeckpeper, Siddharth Singh, Sergey Levine, and Chelsea Finn. RoboNet: Large-scale multi-robot learning. CoRL. 2019.
>
> **A1** Thank you for your valuable suggestions. For the BAIR dataset, we have added `dataloader_bair.py` in the `openstl/datasets/` directory. Due to the limited time for the rebuttal period, we will organize the results of this dataset and upload them to Github. We also plan to include RoboNet in our benchmark in the future.
>
> ***
>
> **Q2** The integration of recurrent-free models and MetaFormers is inspiring. The authors test a wide range of popular Transformer architectures. However, I do not see a consistent trend in the performance between these models. Can the authors provide more analysis here? E.g., if I am a beginner in STL, I'd like to know what temporal module should I choose for a recurrent-free predictor.
>
> **A2** Thanks for your kind suggestion! Different temporal modules indeed perform differently on various datasets. Here we can provide some empirical guidelines for users to reference: If one prioritizes efficiency over performance, ConvMixer stands out due to its fewer parameters, reduced FLOPs, and swift inference. On the other hand, if performance is of the essence, SwinTransformer is preferable for low-frequency data, as showcased by its impressive results on wind component prediction. For high-frequency data, TAU is typically a better option since it has demonstrated stronger temporal modeling capabilities on the Moving MNIST dataset.
>
> ***

---

> > ### Comment · Reviewer_sUSg · 2023-08-17
> >
> > I thank the authors for the response. They address most of my questions. Can the authors also comment on what I posted in the Additional Feedback section of the review? Anyways, I will keep of original rating of Accept.

---

> > > ### Author Response · Authors · 2023-08-17
> > >
> > > Thank you very much for your timely reply. Yes, we can comment here. We are pleased to see some concerns are addressed!

---

### Official Review · Reviewer_jUek · 2023-07-14

**Rating:** 7
**Confidence:** 4
**Correctness:** I think the results are correct.
**Clarity:** Yes.

**Strengths:**

- Spatio-temporal predictive learning is an important problem.
- The source code is released.
- This paper is well organized and clearly written. The technical details are also easy to follow.

**Additional Feedback:**

N/A

**Documentation:**

Yes.

**Opportunities For Improvement:**

- I suggest authors to include more details about the difference of the compared methods.

- Authors should include more SOTAs in this topic [1,2]. If the code is not released, please include a discussion instead.

[1] PastNet: Introducing Physical Inductive Biases for Spatio-temporal Video Prediction, arxiv 23.

[2] Implicit Stacked Autoregressive Model for Video Prediction, arxiv 23

**Relation To Prior Work:**

Yes.

**Summary And Contributions:**

This paper proposes a comprehensive benchmark and open-source toolbox for spatio-temporal predictive learning that supports 14 state-of-the-art recurrent-based and recurrent-free models. The source code is released.

---

> ### Author Response · Authors · 2023-08-17
>
> Dear Reviewer jUek,
>
> Thanks for your insightful and helpful suggestions!
>
> ***
>
> **Q1** I suggest authors to include more details about the difference of the compared methods.
>
> **A1** We acknowledge the importance of delineating the differences between the compared methods to provide a clearer perspective to readers. We summarize the compared methods in the table below.
>
> | Category | Method | Spatial modeling | Temporal modeling |
> |----------|--------|------------------|-------------------|
> | Recurrent-based | ConvLSTM | Conv2D | Conv-LSTM |
> |  | PredNet | Conv2D | ST-LSTM |
> |  | PredRNN | Conv2D | ST-LSTM |
> |  | PredRNN++ | Conv2D | Casual-LSTM |
> |  | MIM | Conv2D | MIM Block |
> |  | E3D-LSTM | Conv3D | E3D-LSTM |
> |  | CrevNet | Conv3D | ST-LSTM |
> |  | PhyDNet | Conv2D | ConvLSTM+PhyCell |
> |  | MAU | Conv2D | MAU |
> |  | PredRNNv2 | Conv2D | ST-LSTM |
> |  | DMVFN | Conv2D | MVFB |
> | Recurrent-free | SimVP | Conv2D | IncepU |
> |  | TAU | Conv2D | TAU |
> |  | SimVPv2 | Conv2D | gSTA |
>
> It can be observed that these methods are very similar in the spatial modeling aspect, with the main difference being the use of different temporal modeling methods.
>
> ***
>
> **Q2** Authors should include more SOTAs in this topic [1,2]. If the code is not released, please include a discussion instead.
>
> [1] PastNet: Introducing Physical Inductive Biases for Spatio-temporal Video Prediction, arxiv 23.
>
> [2] Implicit Stacked Autoregressive Model for Video Prediction, arxiv 23.
>
> **A2** Thanks for your kind reminder. We have added the discussion about these two approaches in our revised manuscript, please refer to the blue color text in Sec 2.3. We will include these two methods in the future plans of OpenSTL and conduct comprehensive experiments to evaluate these approaches.
>
>
> ***

---

> > ### Comment · Reviewer_jUek · 2023-08-21
> > **Thanks for the response.**
> >
> > Thanks for the response. Authors have resolved my concerns.

---

### Official Review · Reviewer_HExp · 2023-07-20
**Review for 843**

**Rating:** 6
**Confidence:** 2
**Correctness:** Yes.
**Clarity:** Yes.

**Strengths:**

- The author has conducted comprehensive experiments, providing valuable references for analyzing existing methods in this field.
- The author has performed a thorough analysis of the results and obtained insightful conclusions.

**Additional Feedback:**

No.

**Documentation:**

I would suggest that the authors may consider providing additional descriptions regarding reproducibility.

**Ethics:**

No.

**Limitations:**

Refer to *Opportunities For Improvement*.

**Opportunities For Improvement:**

I would like to obtain from this paper the proposed OpenSTL in comparison to previous evaluation methods, as a role of a benchmark.

**Relation To Prior Work:**

I find it challenging to identify the novelty of this work and its relation to previous works.

**Summary And Contributions:**

- The author introduces a new benchmark, namely OpenSTL, which covers existing Spatio-Temporal Predictive Learning methods and classifies them into two categories.
- The author conducts detailed experiments on various datasets and offers guidance for future work in this field.

---

> ### Author Response · Authors · 2023-08-17
>
> Dear Reviewer HExp,
>
> Thanks for your constructive comments! We respond to the questions as follows:
>
> ***
>
> **Q1** I would like to obtain from this paper the proposed OpenSTL in comparison to previous evaluation methods, as a role of a benchmark. I find it challenging to identify the novelty of this work and its relation to previous works.
>
> **A1** We are extremely grateful for your valuable suggestions. To the best of our knowledge, our work is the first to systematically evaluate the current state-of-the-art spatio-temporal predictive learning methods. OpenSTL is the first to propose dividing common spatio-temporal predictive learning models into two categories: recurrent-based and recurrent-free models. By introducing a general architecture for recurrent-free models, OpenSTL proposed the inclusion of MetaFormers to extend recurrent-free models, greatly advancing the development of this class of methods.
>
> ***
>
> **Q2** I would suggest that the authors may consider providing additional descriptions regarding reproducibility.
>
> **A2** As indicated in the `install.md` of our Github repository, OpenSTL has provided a conda environment setting file. Users can easily reproduce the environment with the following commands:
>
> ```
> git clone https://github.com/chengtan9907/OpenSTL
> cd OpenSTL
> conda env create -f environment.yml
> conda activate OpenSTL
> python setup.py develop  # or `pip install -e .`
> ```
>
> We have also provided dataset download links, and users can organize the data as follows:
>
> ```
> OpenSTL
> ├── configs
> └── data
>     ├── caltech
>     │   ├── set06
>     │   ├── ...
>     │   ├── set10
>     │   ├── data_cache.npy
>     │   ├── indices_cache.npy
>     ├── human
>     |   ├── images
>     |   ├── test.txt
>     |   ├── train.txt
>     ├── kinetics400
>     │   ├── annotations
>     │   ├── replacement
>     │   ├── test
>     │   ├── train
>     │   ├── val
>     |── kitti_hkl
>     |   ├── sources_test_mini.hkl
>     |   ├── ...
>     |   ├── X_train.hkl
>     │   ├── X_val.hkl
>     |── kth
>     |   ├── boxing
>     |   ├── ...
>     |   ├── walking
>     |── moving_fmnist
>     |   ├── fmnist_test_seq.npy
>     |   ├── train-images-idx3-ubyte.gz
>     |── moving_mnist
>     |   ├── mnist_test_seq.npy
>     |   ├── train-images-idx3-ubyte.gz
>     ├── softmotion30_44k
>     │   ├── test
>     │   ├── train
>     |── taxibj
>     |   ├── dataset.npz
>     |── weather
>     |   ├── 2m_temperature
>     |   ├── ...
>     |── weather_1_40625deg
>     |   ├── 2m_temperature
>     |   ├── ...
> ```
>
> Results can be easily reproduced by running the `tools/train.py` script. For instance, to train the single GPU non-distributed SimVPv2 on the Moving MNIST dataset, the following command line can be used:
>
> ```
> python tools/train.py -d mmnist --lr 1e-3 -c configs/mmnist/simvp/SimVP_gSTA.py --ex_name mmnist_simvp_gsta
> ```
>
> Furthermore, we have provided pre-trained model weights and logs at https://github.com/chengtan9907/OpenSTL/releases. Users can load the pre-trained weights to reproduce the results presented in our manuscript.
>
> ***

---

> > ### Comment · Reviewer_HExp · 2023-08-26
> >
> > The author response has addressed my concerns. I believe this work would benefit STL researchers, and so will raise my rating.

---

### Official Review · Reviewer_rqPr · 2023-07-21
**A comprehensive and insightful benchmark study**

**Rating:** 7
**Confidence:** 4
**Correctness:** The claims are correct.
**Clarity:** The paper is well-structured and easy…

**Strengths:**

1. This work constructs a unified framework for spatial-temporal predictive learning. Various models and modules can be quickly evaluated across datasets and metrics from the well-maintained GitHub repository. It enables the community to evolve their algorithms with minimal effort within such a framework.

2. It conducts a comprehensive benchmarking study on 14 algorithms and 3 representative datasets with 4 metrics. It provides an overview of the recent progress of predictive learning.

3. It provides an insight that recurrent-free models are much more efficient without sacrificing performance, which is very practical for real-world employment. It also finds that meta-former-based models can capture global temporal information, which sheds light on the low-frequency temporal change scenarios.

**Additional Feedback:**

N.A

**Documentation:**

The benchmark is well hosted in the link provided.

**Ethics:**

N.A

**Limitations:**

The same as above.

**Opportunities For Improvement:**

1. When we deploy the system in the real world, we consider not only generalization and efficiency. Another robustness dimension can also be considered in the unified framework for more practical scenarios.

2. This work made an assumption that the competence of recurrent-free models comes from the general arch instead of the specific modules. However, I didn't see any analysis to justify this assumption in the experimental results across datasets. It would be more convincing if there is any evidence, e.g., from a confidence interval aspect.

**Relation To Prior Work:**

It is clear that it bridges the gap with a systematic framework and study.

**Summary And Contributions:**

This work conducts a comprehensive benchmarking study in spatial-temporal predictive learning area. It separates the recent methods into recurrent-based and recurrent-free, and compare their performance and efficiency across model backbones and datasets. Specifically, it categorizes the spatial-temporal-spatial structure into recurrent-free models and uses Meta-formers as the middle temporal modules. It finds that recurrent-free models are competitive in accuracy and much more efficient than recurrent-based models.

---

> ### Author Response · Authors · 2023-08-17
>
> Dear Reviewer rqPr,
>
> Thank you for your thoughtful comments. We hope to address your concerns through the following responses.
>
> ***
>
> **Q1** When we deploy the system in the real world, we consider not only generalization and efficiency. Another robustness dimension can also be considered in the unified framework for more practical scenarios.
>
> **A1** Thanks for your insightful suggestion! The current OpenSTL benchmark primarily focuses on assessing model generalization and efficiency, but expanding it to also systematically evaluate robustness is an important direction for future work. Robustness ensures that the model remains stable and produces consistent results even under adverse conditions or when faced with unforeseen data.
>
> We'll consider incorporating robustness evaluations in the subsequent versions of OpenSTL to provide a more holistic benchmarking platform. We plan to expand on robustness in the future from the following perspectives: (1) Introducing datasets with noise, occlusions, and missing frames; (2) Using techniques such as adversarial training to measure performance in the worst-case scenarios; (3) Designing baseline models with enhanced robustness.
>
> By expanding OpenSTL along these dimensions, we could gain a more comprehensive understanding of model capabilities and limitations for real-world usage. Thank you for the valuable comment!
>
> ***
>
> **Q2** This work made an assumption that the competence of recurrent-free models comes from the general arch instead of the specific modules. However, I didn't see any analysis to justify this assumption in the experimental results across datasets. It would be more convincing if there is any evidence, e.g., from a confidence interval aspect.
>
> **A2** We appreciate the reviewer's comment on the assumption made regarding recurrent-free models. The primary intention behind this assumption was to isolate the inherent advantages of the general architecture from specific modules, thus focusing on broader architectural traits rather than individual components.
>
> In our manuscript, we adopted the general MetaVP architecture and experimented with various MetaFormers as the temporal modules to evaluate this assumption. These recurrent-free models have shown similar results, so we believe this assumption is reasonable.
>
> ***

---

### Comment · Area_Chair_XVgu · 2023-08-17
**Reminder: author-reviewer discussion phase ending soon**

The author-reviewer discussion phase will end on Aug 28th.
@Authors: please write your rebuttal soon. The sooner this is done, the more likely you'll get the reviewers to respond to you in time before the end of the discussion phase.
@Reviewers: please look at the authors' comments once they're available and also other reviewers' comments.

Thank you.

Area Chair

---

### Decision · Program_Chairs · 2023-09-22

**Decision:**

Accept (Poster)

**Comment:**

The paper presents a new and comprehensive spatio-temporal learning in two categories - recurrent-based and recurrent-free, and tested on diverse datasets including driving scenes, traffic flow, and weather data.
The authors provide a modular and extensible framework, easily customisable.
We ask the authors to add the robustness aspect of the evaluation requested by by reviewer rqPr in the final camera ready of the paper.